# The impact of the management strategies for patients with subclinical hypothyroidism on long-term clinical outcomes: An umbrella review

**Brenda S. Bauer** [ID]*, **Amaya Azcoaga-Lorenzo, Utkarsh Agrawal** [ID]**, Adeniyi Francis Fagbamigbe, Colin McCowan** [ID]

Population and Behavioural Science Division, School of Medicine, University of St Andrews, St Andrews, Scotland, United Kingdom

* bsb1@st-andrews.ac.uk

## Abstract

### Aim

This umbrella review summarises and compares synthesised evidence on the impact of subclinical hypothyroidism and its management on long-term clinical outcomes.

### Methods

We conducted comprehensive searches on MEDLINE, EMBASE, Scopus, Web of Science, Cochrane Database of Systematic Reviews, JBI Evidence Synthesis, the PROSPERO register, Epistemonikos Database and PDQ Evidence from inception to February and July 2021 using keywords on subclinical hypothyroidism, treatment with levothyroxine, monitoring and primary outcomes (all-cause mortality, cardiovascular events, stroke, frailty fractures and quality of life). Only systematic reviews and meta-analyses on adult patient populations were considered. Study selection, data extraction and quality appraisal using AMSTAR-2 were done independently by two reviewers and discrepancies were resolved through discussion. Overlap across the selected reviews was also assessed, followed by a narrative synthesis of findings.

### Results

A total of 763 studies were identified from literature searches; 20 reviews met inclusion criteria. Methodological quality ratings were high (n = 8), moderate (n = 7), and low (n = 5), but no reviews were excluded on this basis. Though there was slight overlap across all reviews, some pairwise comparisons had high corrected covered area scores. Compared to euthyroidism, untreated subclinical hypothyroidism was associated with a higher risk of cardiovascular events or death if Thyroid Stimulating Hormone was above 10mIU/L at baseline. Treatment was associated with a lower risk of death from all causes for patients younger than 70 years and possibly better cognitive and quality of life scores than untreated individuals. Evidence on the risk of strokes and fractures was inconclusive.

**Data Availability Statement:** All relevant data are within the manuscript and its Supporting Information files.

**Funding:** This umbrella review was conducted as part of a PhD project (BSB) funded by the School of Medicine, University of St Andrews, Scotland, United Kingdom. Publication fees were funded under an Open Access membership agreement between the University of St Andrews Library and PLOS. The funders had no role in study design, data collection and analysis, decision to publish, or preparation of the manuscript.

**Competing interests:** The authors have declared that no competing interests exist.

## Conclusion

In the long term, treatment of subclinical hypothyroidism may be beneficial for some patient groups. However, the findings of this review are negatively impacted by the relative sparseness and poor quality of available evidence. Additional large and adequately powered studies are needed to investigate this topic further.

## Systematic review registration

PROSPERO (CRD42021235172)

## Introduction

Subclinical hypothyroidism (SCH) is characterised by elevated Thyroid Stimulating Hormone (TSH) levels in contrast to free thyroid hormone–usually thyroxine/T4 –within the reference range [1–3]. The leading cause of SCH is Hashimoto's thyroiditis, a chronic autoimmune disorder that affects more women than men [2]. Prevalence varies worldwide but has been estimated to be between 4% to 9%, increasing with age to more than 20% for women over 60 years old [2, 4]. This broad range can also be explained by differences in race, dietary intake of iodine and diagnostic cut-offs for SCH [2]. Patients usually exhibit few, if any, symptoms, so it is common for SCH to be detected incidentally from a routine blood test panel [5, 6].

The reference ranges used for measurements of thyroid hormones and TSH vary between laboratories because they are highly dependent on the reference population [7]. Nonetheless, a distinction is sometimes made between mild and severe SCH with a TSH measurement of 10mIU/L as the cut-off [8, 9]. Approximately 60% of cases with mild SCH revert to normal TSH levels over time [3, 9]. Furthermore, depending on the initial severity of their condition, female SCH patients and those that are antithyroid peroxidase antibody-positive are more likely to develop overt hypothyroidism [4]. This progression occurs in around 2–4% of cases per year [2, 3].

Measurement of TSH and thyroid hormone levels is achieved through thyroid function tests (TFTs) which are frequently ordered unnecessarily without medical indications [10–12]. One of the main pitfalls that could likely result from inappropriate TFTs is that more asymptomatic patients are diagnosed as having SCH. Following diagnosis, there are two options for the management of SCH, thyroid replacement therapy with levothyroxine or follow-up without prescribing medication [2, 3]. Even so, a reasonable expectation for the latter is that treatment would be initiated in the event of a patient's worsening state, provided that the progression can be attributed to SCH. Regardless of the strategy, patients require periodic blood tests to monitor TSH levels for increasing severity or improvement of SCH [1, 3, 13].

The management of SCH is controversial–there has been no definitive evidence on the benefits of replacing thyroid hormones, especially the long-term clinical consequences. This is partly because few adequately powered randomised trials have investigated this topic [4, 14], the most notable being the Thyroid Hormone Replacement for Untreated Older Adults with Subclinical Hypothyroidism (TRUST) trial [15]. The TRUST Study Group reported that in their trial with 737 adults over 65 years old, levothyroxine treatment did not improve patient symptoms nor lower the risk of cardiovascular events and fractures [15]. On the other hand, other smaller trials and observational studies have linked treatment of SCH to improved patient outcomes [2, 4, 8]. Also, current UK clinical guidelines for the management of SCH

differ in their recommendations for treatment thresholds and exclusions [14, 16]; these differences can be directly ascribed to inconsistencies in the existing evidence base.

Based on a systematic review by Feller et al. [17], a clinical guideline panel found no evidence to recommend thyroid hormone replacement for SCH patients, except for those with TSH levels above 20mIU/L and women that are pregnant or trying to conceive [14]. For the outcomes they considered, for example, quality of life, cognitive function and cardiovascular events, the panel found no vital difference between treated and untreated groups, irrespective of patient age. Moreover, they noted the issue of practicality regarding medication–patients require long-term treatment and follow-up and even risk developing hyperthyroidism in case of overuse [14]. In contrast, the latest NICE guidance recommendation is for physicians to consider treating adults with TSH of $\geq 10$ mIU/L to improve SCH patient outcomes [16]. The reviewing committee found little evidence on SCH treatment but emphasised that additional factors–such as the presence of symptoms–should be considered, over and above TSH levels [13].

Therefore, an umbrella review was performed to collate and compare existing literature on the long-term effects of SCH treatment and follow-up with no medication. Umbrella reviews–also called overviews, meta-reviews or reviews of reviews [18]–are ideally suited to exploring discrepancies in the literature by allowing for a broader scope of inquiry than a typical systematic review [18, 19]. The review questions of interest were: (i) what is the impact of levothyroxine treatment on patient outcomes in subclinical hypothyroidism? and (ii) what is the impact of monitoring without treatment on clinical outcomes for patients with subclinical hypothyroidism? Rather than restrict the focus of this overview to a direct comparison of these management strategies, we sought also to identify what is known for either option.

## Methods

To a large extent, umbrella reviews are conducted similarly to typical systematic reviews. However, the critical difference between these methods is that the former use existing systematic reviews and meta-analyses as the units of synthesis [18]. These will subsequently be referred to as 'primary reviews' in this paper, in contrast to 'primary studies', the empirical studies included in the systematic reviews. The protocol for this overview was registered on PROSPERO (CRD42021235172) and the methods followed have previously been described in detail [20]; there were no deviations from the registered protocol. The reporting of this overview follows a checklist developed for overviews of systematic reviews based on recommendations from existing guidelines [21].

### Search strategy

Comprehensive searches were performed on multiple databases from inception to February 2021, namely MEDLINE, EMBASE, Scopus, Web of Science, Cochrane Database of Systematic Reviews, JBI Evidence Synthesis, the PROSPERO register, Epistemonikos Database and PDQ Evidence. The key search terms used included 'subclinical hypothyroidism', 'monitoring', 'treatment' and 'levothyroxine', in both free text and subject headings; the search syntax was modified to match the different databases. The searches were updated in July 2021, during the latter stages of data extraction, to identify any systematic reviews and meta-analyses that had been published as the review was in progress. The MEDLINE search strategy is provided in S1 Appendix. No additional date or language filters were applied. Grey literature searches were also performed, and the reference lists of eligible studies were scanned to identify other potentially relevant primary reviews.

## Eligibility criteria

The following eligibility criteria were considered to determine inclusion in this umbrella review. Only systematic reviews and meta-analyses on SCH, either in part or as a whole, were considered, irrespective of whether the primary review included randomised trials or observational studies. Generally, all primary reviews had to report on the clinical outcomes of adult patients (>18 years old) with SCH, regardless of the diagnostic thresholds that were initially applied, for example, reference ranges for TSH and thyroid hormones. Patients with overt hypothyroidism were excluded, as were children and pregnant women, whose thyroid hormone requirements differ from the rest of the population.

The two eligible interventions were: (i) treatment with levothyroxine, and (ii) patient follow-up without medication. Another essential requirement was that the treatment status of patients was reported in the systematic reviews, such that it would be possible to distinguish between treated and untreated groups. There were no additional restrictions on study comparators and settings.

The primary outcomes of interest were all-cause mortality, defined as the death of patients with SCH, irrespective of the cause, at least 12 months from baseline or the start of follow-up; cardio- and cerebrovascular outcomes such as heart failure, arrhythmias, stroke, peripheral vascular disease, coronary heart disease; quality of life as measured using suitable instruments (or otherwise described as 'symptoms' particularly in older publications); and, frailty fractures, defined as fractures resulting from low-impact trauma, usually due to pre-existing disease. Other long-term clinical outcomes reported in the included systematic reviews, for example, cognitive function, were considered secondary outcomes.

## Study selection

References retrieved from the searches were imported into Covidence (www.covidence.org/) and initially screened in duplicate for eligibility by title and abstract. After that, the full texts of selected primary reviews were obtained and read independently by pairs of reviewers who assessed each paper against the selection criteria. When needed, primary review authors were contacted to provide additional information.

## Data extraction

A data extraction form was developed on Covidence and piloted by two reviewers. This form was used to extract information on citation details, primary study selection criteria, search parameters, selection and quality assessment methods and primary review findings relating to the outcomes of interest for this umbrella review. Where provided, effect estimates were extracted alongside their 95% confidence intervals and the treatment status of the assessed group(s). This process was done by two reviewers working independently, and discrepancies in the extracted data were resolved through discussion to reach a consensus.

**Table 1. AMSTAR-2 overall confidence ratings (from Shea et al. [22]).**

| Rating | Interpretation |
| --- | --- |
| High | ≤1 non-critical weakness |
| Moderate | >1 non-critical weakness |
| Low | 1 critical flaw +/- non-critical weaknesses |
| Critically low | >1 critical flaw +/- non-critical weaknesses |

## Quality appraisal

The quality of the selected systematic reviews was independently assessed by two reviewers using the Assessment of Multiple Systematic Reviews (AMSTAR-2) tool [22], an instrument with 16 questions on the methodological quality of systematic reviews. These questions include whether a comprehensive literature search was conducted, justification for excluding studies, and risk of bias assessments for included studies (S2 Appendix). Overall ratings and their meanings are shown in Table 1. AMSTAR-2 was chosen over the risk of bias in systematic reviews (ROBIS) tool [23] because while both assess strongly related aspects (methodological quality vs risk of bias), the former is advantageous for inter-rater reliability and usability [24–26]. Disagreements between the reviewers were similarly resolved through discussion.

It has been suggested that GRADE criteria can be applied to systematic reviews [27]. However, this approach was initially designed for empirical studies, hence the paucity of relevant guidance on how best to achieve this [18, 28]. Therefore, we did not perform any secondary GRADE assessments on the included primary reviews but extracted any reported quality ratings.

## Assessing overlap

One of the unique challenges in conducting an umbrella review is overlap–the inclusion of the same primary study or trial in more than one selected systematic review or meta-analysis [29, 30]. Any subsequent synthesis of more than one of these primary reviews would result in 'double-counting' and biased findings because the contribution of a subset of the data would have been multiplied by some factor [30, 31]. The proposed methods for dealing with overlap are: (i) selecting only the most recent systematic review or the one with the largest number of studies, (ii) selecting only the primary review of the highest quality, or (iii) including all primary reviews but evaluating the amount of overlap [18, 30, 31].

To assess overlap between the included primary reviews, we calculated the corrected covered area (CCA) using the formula described by Pieper et al. [29]:

$$CCA\ (Corrected\ CA) = \frac{N - r}{rc - r}$$

where N–number of included primary studies in selected reviews

r–number of index publications

c–number of included primary reviews

A matrix of the included systematic reviews and their primary studies was created to identify the numerators and denominators shown above. CCA is interpreted in banded thresholds: 5% or less indicates slight overlap, 6% to 10% shows moderate overlap, 11% to 15% for high overlap and values greater than 15% indicate very high overlap [29].

## Synthesis of results

A narrative synthesis of results was performed due to high levels of study overlap and considerable heterogeneity in primary review inclusion criteria and reported outcomes. Summaries of the included primary reviews are presented below in tabular form alongside corresponding effect estimates such as odds and hazard ratios (where reported). Of note, no further re-analysis of empirical study data was performed, as previously stated in the umbrella review protocol [20]. The extracted data were grouped according to the clinical outcome of interest, regardless of the degree of overlap among sets of studies. Moreover, because there is currently no agreed-upon solution for the issue of low-quality systematic reviews in overviews [18, 26], all selected papers were included in the narrative synthesis.

## Results

In total, 763 records were retrieved from the initial and updated searches. After screening by title, abstract and full-text, 20 syntheses were selected for inclusion in this umbrella review. Notably, two otherwise eligible primary reviews were excluded based on all their included studies having been used in later publications by the same authors [32, 33]. Authors were unable to provide further information for two other publications [34, 35]. One item of grey literature, a systematic evidence review commissioned by a government agency for healthcare research, was included [36]. A list of the systematic reviews that were excluded after reading full texts is provided in S3 Appendix. The PRISMA flowchart [37] showing the stages of study selection is presented in Fig 1.

The characteristics of the selected primary reviews are shown in Table 2. Of the 20 included syntheses, four were systematic reviews [36, 38–40], five were published as combined systematic reviews and meta-analyses [17, 41–44], six were labelled meta-analyses [45–50], and five were individual participant data analyses [51–55]. The majority were published earlier than the TRUST trial [15], with only seven primary reviews published during or after 2017 [17, 38, 41–43, 50, 51]. Most of the primary reviews synthesised observational data, although three papers only included RCTs [17, 36, 40]. Generally, SCH was defined using similar TSH thresholds (> 4.5 mIU/L) and normal T4, but some studies subdivided this further into degrees of SCH or TSH elevation e.g., mild vs moderate (Table 2).

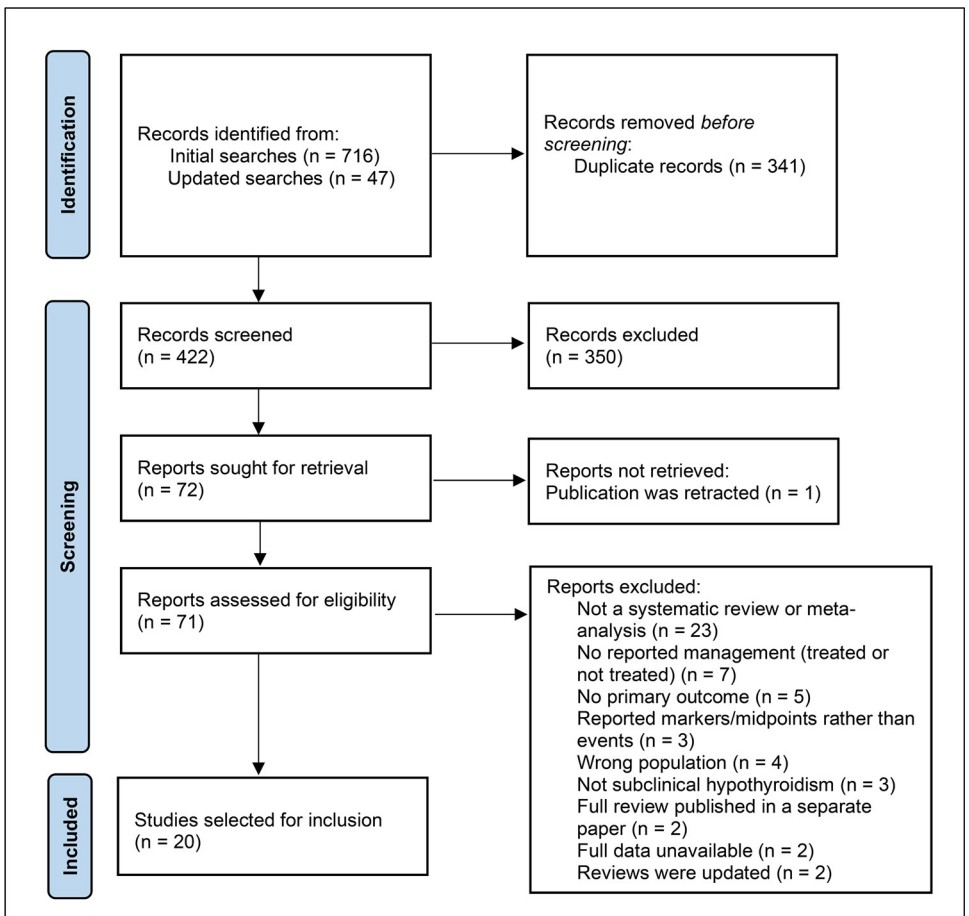

**Fig 1. PRISMA flowchart showing study selection (adapted from Page et al. [37]).**

**Table 2. Characteristics of the included systematic reviews and meta-analyses.**

| Study | Design | Study aim | Included studies | Definition of SCH | SCH patients (%) | Summary of findings | AMSTAR-2 Overall confidence |
|---|---|---|---|---|---|---|---|
| Baumgartner et al. (2017) [51] | IPD | To examine the risk of AF in individuals with thyroid function within the normal range and SCH | 11 cohort studies (IPD) | TSH level between 4.5 and 19.9 mIU/L with fT4 levels in the reference range | 1958 (6.5) | The reviewers found no link between SCH and the risk of AF; this was the same for individuals with TSH levels within the normal range. | High |
| Blum et al. (2015) [45] | MA | To assess the association of subclinical thyroid dysfunction with fractures | 13 cohort studies | TSH level of 4.50 to 19.99 mIU/L with normal FT4 levels | 4092 (5.8) | There was no observed association between SCH and fracture risk. | Moderate |
| Chaker et al. (2015) [52] | IPD | To evaluate the association between SCH and stroke | 17 cohort studies | TSH levels of 4.5 to 19.9 mIU/L with normal T4 levels | 3451 (7.3) | There was no overall increase in the risk of stroke events and fatal stroke in patients with SCH than euthyroid patients, except for patients younger than 65 years. | Moderate |
| Collet et al. (2014) [53] | IPD | To compare the risks of CHD mortality and events associated with SCH by thyroid antibody status | 6 cohort studies | TSH 4.5 to 19.9 mIU/L and normal T4 level | 1691 (4.4) | Thyroid antibodies were found to have no effect on CHD events and mortality though SCH patients with higher TSH levels were generally at higher risk of developing these outcomes. | Moderate |
| Dhital et al. (2017) [41] | SR + MA | To look at the association between thyroid function profile and outcomes after acute ischemic stroke | 12 cohort studies | Elevated TSH and normal fT4 (study-specific cut-offs) | Unclear | SCH was associated with better functional outcomes after acute ischemic stroke, but this depended on the initial levels of free T3. | Low |
| Feller et al. (2018) [17] | SR + MA | To examine the association of THT with quality of life and thyroid-related symptoms in adults with SCH | 21 RCTS | Thyrotropin and free thyroxine levels above and within centre-specific reference ranges, respectively | 2192 (100) | There was no association between treatment of SCH and improving thyroid-related symptoms and quality of life (primary outcomes) or cognitive function, depressive symptoms and the other secondary outcomes. | High |
| Gencer et al. (2012) [54] | IPD | To clarify the association between subclinical thyroid dysfunction and HF events | 6 cohort studies | TSH level of 4.5 to 19.9 mIU/L with normal FT4 levels | 2068 (8.1) | Patients with TSH levels higher than 10mIU/L faced a significantly higher risk of HF events. | Moderate |
| Helfand (2004) [36] | SR | To evaluate the benefits of screening for subclinical thyroid dysfunction | 8 RCTs | Elevated TSH and normal T4 | Unclear | Evidence of an association between treatment and reduced symptoms was demonstrated only for SCH patients with TSH >10 mIU/L and those with a history of Graves' disease. | Low |
| Peng et al. (2021) [42] | SR + MA | To investigate whether THT is associated with decreased mortality in adults with SCH | 2 RCTs and 5 cohort studies | Grade 1 (TSH level 5.0–10 mIU/L); Grade 2 (TSH level >10 mIU/L) with free thyroxine level within the reference range | 21055*3 (100) | Treatment was found to benefit SCH patients younger than 65 years; all-cause mortality decreased by 50%, and cardiovascular mortality decreased by 46%. However, the same did not apply to patients older than 65 years. There was also no overall benefit of treatment on mortality. | High |

(*Continued*)

**Table 2.** (Continued)

| Study | Design | Study aim | Included studies | Definition of SCH | SCH patients (%) | Summary of findings | AMSTAR-2 Overall confidence |
|---|---|---|---|---|---|---|---|
| Razvi et al. (2008) [46] | MA | To examine the influence of age and gender on IHD and mortality in SCH | 15 cohort studies | Mild SCH—TSH levels < 10 mIU/L | 2,531 (8.7) | The overall incidence of IHD and mortality was not significantly higher for patients with SCH, but IHD prevalence was found to be significantly elevated for patients younger than 65 years. | High |
| Reyes Domingo et al. (2019) [38] | SR | To synthesize the evidence on the effects of screening and subsequent treatment for thyroid dysfunction | 5 RCTs and 3 cohort studies* 2 | Study-specific | Unclear | Evidence was found linking treatment for SCH with reduced all-cause mortality for patients younger than 65 years, but it was determined to be of low quality. | High |
| Rodondi et al. (2006) [47] | MA | To determine whether SCH is associated with an increased risk for CHD | 5 cohort, 6 cross-sectional and 3 case-control studies | Elevated TSH and a normal T4 (no pre-specified cut-offs) | 1409 (10.8) | Compared to euthyroid patients, CHD was 1.6 times more likely in patients with SCH; this association was constant throughout the included studies but less pronounced in the prospective cohorts. | High |
| Rodondi et al. (2010) [55] | IPD | To assess the risks of CHD and total mortality for adults with SCH | 11 cohort studies | Serum TSH level of 4.5 mIU/L or greater to less than 20 mIU/L, with a normal T4 concentration | 3450 (6.2) | SCH patients with TSH levels higher than 10mIU/L had a significantly higher risk of CHD events and mortality than euthyroid patients. | High |
| Rugge et al. (2015) [39] | SR | To assess the benefits and harms of screening and treatment of subclinical and undiagnosed overt hypothyroidism and hyperthyroidism in adults* | 13 RCTs and 1 cohort study | 4.5–10.0 mIU/L (mildly elevated) or ≥10 mIU/L (markedly elevated) TSH levels with normal thyroxine | Unclear | Reviewers found a potential association between SCH and cardiovascular disease but inconclusive evidence that treatment would be beneficial; SCH treatment was also not associated with improved cognitive function or quality of life. | Moderate |
| Singh et al. (2008) [48] | MA | To compare the relative risk for incident CHD events, cardiovascular-related and total mortality associated with subclinical thyroid abnormalities | 6 cohort studies | Serum TSH above 4.0–5.0 mIU/L with normal free T4 (range 0.7–1.8 ng/dL) | 1365 (10.2) | SCH was linked to a significant risk of CHD at baseline and both CHD and cardiovascular mortality during follow-up. On the other hand, all-cause mortality was not found to be increased with SCH. | Low |
| Sun et al. (2017) [43] | SR + MA | To explore the relationship between subclinical thyroid dysfunction and the risk of cardiovascular outcomes | 16 cohort studies | TSH levels >3.6 to 6 mIU/L (study-specific) | 5178 (7.2) | There was a significantly higher risk of CHD and cardiovascular mortality for SCH patients younger than 65 years, but the same effect was not observed for patients older than 80 years. A slightly higher risk of AF and HF was also associated with SCH. | Moderate |

(*Continued*)

**Table 2.** (Continued)

| Study | Design | Study aim | Included studies | Definition of SCH | SCH patients (%) | Summary of findings | AMSTAR-2 Overall confidence |
|---|---|---|---|---|---|---|---|
| Villar et al. (2007) [40] | SR | To assess the effects of thyroid hormone replacement for SCH | 12 RCTs | TSH level above the upper limit of the reference range with normal values of total T4 or free T4 (FT4), with or without T3 or free T3 (FT3) measurements | 350 (100) | It was not possible to assess the benefits of SCH treatment on reducing cardiovascular mortality. However, there was also no significant impact of levothyroxine on health-related quality of life and symptoms. | High |
| Wirth et al. (2014) [44] | SR + MA | To assess the risk for hip and non-spine fractures associated with subclinical thyroid dysfunction | 7 cohort studies | TSH level greater than 4.5 to 20.0 mIU/L and an FT4 level in the reference range | Unclear | No association between SCH and fracture risk was found, but the reviewers could not assess the effects of treatment vs no treatment due to insufficient data. | Moderate |
| Yan et al. (2016) [49] | MA | To identify the relationship between subclinical thyroid dysfunction and the risk of fracture | 5 cohort studies | TSH level greater than 4.0 to 5.5 mIU/L (study-specific) | 2580 (0.9) | A link between SCH and higher fracture risk was not found, but the reviewers acknowledge that they had limited data. | Low |
| Yang et al. (2019) [50] | MA | To assess the association between subclinical thyroid dysfunction and the clinical outcomes of HF patients | 14 cohort studies | Elevated TSH values in the presence of normal FT4 values | 2308 (10.9) | Both adjusted and unadjusted analyses showed a significantly higher risk of all-cause mortality and cardiovascular death associated with SCH for patients with heart failure. | Low |

THT—Thyroid Hormone Therapy; SR–Systematic Review; SR + MA–Systematic Review and Meta-Analysis; MA–Meta-analysis; IPD–Individual Participant Data analysis; SCH–Subclinical Hypothyroidism; CHD–Coronary Heart Disease; AF–Atrial Fibrillation; RCT–Randomised Controlled Trial; HF–Heart Failure; IHD–Ischaemic Heart Disease; Thyroxine–T4, fT4, thyroid hormone; Thyrotropin–Thyroid Stimulating Hormone (TSH)

*this was an update to Helfand et al. [36], but because the searches did not overlap, this was considered a separate review.

*2 only for the relevant research question on clinical outcomes for SCH.

*3 the authors report potential overlap between the studies; hence the estimate may be incorrect.

Coverage of the primary outcomes was good, given that all the outcomes of interest were reported in at least two publications. However, it is crucial to note that most relevant results were obtained via subgroup or sensitivity analyses in the primary reviews. As such, they were not necessarily representative of the overall findings shown in Table 2.

## All-cause mortality

Seven publications reported findings on all-cause mortality; of these, three primary reviews compared rates between treated and untreated patients [38, 39, 42], three compared untreated and euthyroid individuals [43, 46, 48], and one compared both treated and untreated SCH groups with euthyroid participants [50]. There was no statistically significant difference in the overall numbers of deaths from all causes for patients with SCH between those who were and were not on treatment (Table 3).

Taking age into account, lower estimates of all-cause mortality were reported for patients younger than 70 years on treatment (RR 0.50, 95% CI 0.29 to 0.85 [42]; HR 0.36, 95% CI 0.19 to 0.66 [38, 39]). However, these estimates were based on one study in Rugge et al. [39], and another paper rated the same evidence as being of very low certainty [38]. On the other hand, older patient groups demonstrated no significant association between levothyroxine treatment and all-cause mortality (Table 3).

**Table 3. Review findings on all-cause mortality.**

| Study | Outcome | Treatment status | Comparator | Effect estimate (95% CI) |
|---|---|---|---|---|
| Peng et al. (2021) [42] | All-cause mortality | Treated | Untreated | RR 0.95 (0.75–1.22) |
| | All-cause mortality; age <65–70 years | | | RR 0.50 (0.29–0.85) |
| | All-cause mortality; age > = 65–70 years | | | RR 1.08 (0.91–1.28) |
| Reyes Domingo et al. (2019) [38] | All-cause mortality; adults (>18 years) | Treated | Untreated | HR 1.91 (0.65–5.60) |
| | All-cause mortality; adults (<65 or <70 years) | | | IRR 0.63 (0.40–0.99 |
| | | | | HR 0.36 (0.19–0.66) |
| | All-cause mortality; adults (>65 years) | | | HR 1.91 (0.65–5.60) |
| | All-cause mortality; females | | | IRR 0.99 (0.85–1.16) |
| | | | | 1.08 (0.80–1.48) |
| | All-cause mortality; males | | | IRR 1.24 (0.89–1.16) |
| | | | | 1.43 (0.87–2.34) |
| Rugge et al. (2015) [39] | All-cause mortality; 40–70 years | Treated | Untreated | HR 0.36 (0.19–0.66) |
| | All-cause mortality; >70 years | | | HR 0.71 (0.56–1.08) |
| Yang et al. (2019) [50] | All-cause mortality | Untreated | Euthyroid | HR 1.48 (1.29–1.70) |
| | | Treated | Euthyroid | HR 1.48 (1.14–1.94) |
| Razvi et al. (2008) [46] | IHD/all-cause mortality; <65 years | Untreated | Euthyroid | OR 1.32 (0.95–1.83) |
| | IHD/all-cause mortality; > 65 years | | | OR 0.87 (0.51–1.45) |
| Sun et al. (2017) [43] | Total mortality | Untreated | Euthyroid | RR 1.01 (0.90–1.15) |
| Singh et al. (2008) [48] | All-cause mortality | Untreated | Euthyroid | RR 1.115 (0.990–1.255) |

HR–Hazard Ratio; RR–Relative Risk; IRR–Incident rate Ratio; IHD–Ischemic Heart Disease; OR–Odds Ratio.

Only one of four comparisons of all-cause mortality between untreated and euthyroid study participants was statistically significant (HR 1.48, 95% CI 1.29 to 1.70) [50]. The same review found that death was more likely among SCH patients on treatment than in euthyroid persons (HR 1.48, 95% CI 1.14 to 1.94) [50]. However, the population of interest for this review all had heart failure, thereby limiting the generalisability of these findings to other SCH patients.

## Cardiovascular outcomes

Cardiovascular outcomes were the most extensively reported outcomes of interest across the included reviews (n = 13) as shown in Table 4. No difference was found in the number of incident atrial fibrillation events between untreated persons and euthyroid controls, irrespective of age and TSH level [51]. Similarly, the difference between treated and untreated SCH patients was not statistically significant, though notably, the evidence was rated as being of very low to moderate certainty [38].

Compared to euthyroidism, untreated SCH was significantly associated with a higher likelihood of CHD and heart failure if patients had TSH levels above 10mIU/L (HR 2.17, 95% CI 1.19 to 3.93) [55], (HR 2.37, 95% CI 1.59 to 3.54) [54] or were thyroid peroxidase antibody-negative (HR 1.25, 95% CI 1.06 to 1.47); HR (3.76, 95% CI 1.77 to 8.01) [53]. It was also reported that untreated SCH was associated with higher odds of ischaemic heart disease (OR 1.58, 95% CI 1.07 to 2.35) [46] and a higher risk of developing coronary heart disease during follow-up (RR 1.188, 95% CI 1.024 to 1.379) [48] than euthyroid participants. However, one primary review found that incident CHD was not associated with untreated SCH [43]–this difference may have resulted from the reviewers' decision to restrict the inclusion of primary studies based on quality appraisal scores.

**Table 4. Reported primary review findings on cardiovascular outcomes.**

| Study | Outcome | Treatment status | Comparator | Effect estimates (95% CI) |
|-------|---------|------------------|------------|---------------------------|
| Baumgartner et al. (2017) [51] | Atrial fibrillation | Untreated | Euthyroid (TSH 3.50–4.49 mIU/L) | For TSH 4.5–6.9 mIU/L: HR 0.87 (0.66–1.16) |
| | | | | For TSH 7.0–9.9 mIU/L: HR 1.22 (0.78–1.92) |
| | | | | For TSH 10.0–19.9 mIU/L: HR 1.56 (0.84–2.90) |
| Reyes Domingo et al. (2019) [38] | Atrial fibrillation; adults (>18y) | Treated | Untreated | HR 0.80 (0.35–1.80) |
| | Atrial fibrillation; adults (<65 or <70) | | | HR 0.76 (0.26–1.73) |
| | Atrial fibrillation; adults (>65y) | | | HR 0.80 (0.35–1.80) |
| Collet et al. (2014) [53] | CHD events | Untreated | Euthyroid | SH With -ve TPOAb HR 1.25 (1.06–1.47) |
| | | | | SH With +ve TPOAb HR 1.12 (0.88–1.41) |
| | | | | SH with TSH ≥10.0 mIU/L and neg. TPOAb HR 3.76 (1.77–8.01) |
| | | | | SH with TSH ≥10.0 mIU/L and pos. TPOAb HR 1.19 (0.61–2.32) |
| Rodondi et al. (2006) [47] | CHD | Untreated | Euthyroid | OR 2.06 (1.36–3.14) |
| Rodondi et al. (2010) [55] | CHD | Untreated | Euthyroid | For TSH 4.5–19.99 mIU/L: HR 1.17 (0.91–1.50) |
| | | | | For TSH 10–19.99 mIU/L: HR 2.17 (1.19–3.93) |
| Sun et al. (2017) [43] | CHD | Untreated | Euthyroid | RR 1.02 (0.92–1.14) |
| Singh et al. (2008) [48] | CHD (during follow-up) | Untreated | Euthyroid | RR 1.188 (1.024–1.379) |
| Gencer et al. (2012) [54] | Heart failure events; TSH 4.5–19.9 mIU/L | Untreated | Euthyroid | HR 1.26 (0.93–1.69) |
| | Heart failure events; TSH 10.0–19.9 mIU/L | | | HR 2.37 (1.59–3.54) |
| Razvi et al. (2008) [46] | IHD incidence; < 65 yrs | Untreated | Euthyroid | OR 1.58 (1.07–2.35) |
| | IHD incidence; > 65 yrs | | | N/P |
| Rugge et al. (2015) [39] | IHD; 40–70 yrs | Treated | Untreated | HR 0.61 (0.39–0.95) |
| | IHD; >70 yrs | | | HR 0.99 (0.59–1.33) |
| Reyes Domingo et al. (2019) [38] | Fatal and non-fatal cardiovascular events (not AF); adults (>18y) | Treated | Untreated | HR 0.89 (0.47–1.69) |
| | Fatal and non-fatal cardiovascular events (not AF); adults (<65 or <70) | | | HR 0.61 (0.39–0.95) |
| | | | | HR 1.03 (0.51–2.13) |
| | | | | IRR 1.11 (0.61–2.02) |
| | Fatal and non-fatal cardiovascular events (not AF); adults (>65y) | | | HR 0.89 (0.47–1.69) |
| | Fatal and non-fatal cardiovascular events (not AF); females | | | IRR 0.99 (0.70–1.38) |
| | | | | 0.99 (0.70–1.40) |
| | Fatal and non-fatal cardiovascular events (not AF); males | | | IRR 1.41 (0.83–2.40) |
| | | | | 1.36 (0.79–2.35) |

CHD–Coronary heart disease; HR–Hazard Ratio; RR–Relative Risk; IRR–Incident rate Ratio; IHD–Ischemic Heart Disease; OR–Odds Ratio.

SCH patients receiving treatment and younger than 70 years were significantly less likely to develop IHD than untreated individuals (HR 0.61, 95% CI 0.39 to 0.95) [38]. However, these findings were based on a single empirical study in both primary reviews, in which it was reported that the GRADE rating for this evidence was very low. A similar association was not found for patients older than 70 years nor subgroups based on sex [38].

**Table 5. Reported primary review findings on cardiovascular mortality.**

| Study | Outcome | Treatment status | Comparator | Effect estimates (95% CI) |
|---|---|---|---|---|
| Peng et al. (2021) [42] | Cardiovascular mortality | Treated | Untreated | RR 0.99 (0.82–1.20) |
| | Cardiovascular mortality; age <65–70 years | | | RR 0.54 (0.37–0.80) |
| | Cardiovascular mortality; age > = 65–70 years | | | RR 1.05 (0.87–1.27) |
| Reyes Domingo et al. (2019) [38] | Cardiovascular deaths; adults (>18y) | Treated | Untreated | OR 2.01 (0.18–22.27) |
| | Cardiovascular deaths; adults (<65 or <70) | | | HR 0.54 (0.37–0.92) IRR 0.55 (0.25–1.20) |
| | Cardiovascular deaths; adults (>65y) | | | OR 2.01 (0.18–22.27) |
| | Cardiovascular deaths; females | | | IRR 0.96 (0.77–1.21) |
| | Cardiovascular deaths; males | | | IRR 1.32 (0.83–2.08) |
| Rugge et al. (2015) [39] | Cardiovascular deaths (40–70 years) | Treated | Untreated | HR 0.54 (0.37–0.92) |
| Collet et al. (2014) [53] | CHD mortality | Untreated | Euthyroid | SH With -ve TPOAb HR 1.34 (1.07–1.69) |
| | | | | SH With +ve TPOAb HR 1.28 (0.94–1.72) |
| | | | | SH with TSH ≥10.0 mIU/L and negative TPOAb HR 1.95 (0.76–4.98) |
| | | | | SH with ≥10.0 mIU/L and positive TPOAb HR 1.92 (1.09–3.36) |
| Rodondi et al. (2010) [55] | CHD mortality | Untreated | Euthyroid | For TSH 4.5–19.99 mIU/L: HR 1.25 (1.04–1.51) |
| | | | | For TSH 10–19.99 mIU/L: HR 1.85 (1.13–3.05) |
| Sun et al. (2017) [43] | Cardiovascular mortality | Untreated | Euthyroid | RR 0.86 (0.56–1.32) |
| Singh et al. (2008) [48] | Cardiovascular mortality | Untreated | Euthyroid | RR 1.278 (1.023–1.597) |
| Yang et al. (2019) [50] | Cardiac death and/or hospitalization | Untreated | Euthyroid | HR 1.32 (1.08–1.60) |
| | | Treated | | HR 1.36 (1.12–1.66) |

CHD–Coronary Heart Disease; HR–Hazard Ratio; RR–Relative Risk; IRR–Incident rate Ratio; OR–Odds Ratio.

We distinguished between all-cause mortality and cardiovascular mortality (n = 8), which primary review authors defined as deaths arising from cardiovascular diseases (Table 5). Considering treated vs untreated SCH, an association between treatment and cardiovascular death was found only for adult patients younger than 65–70 years [38, 39, 42]. Stratifying the results by sex did not yield statistically significant findings. In addition, whereas the risk of cardiovascular mortality was found to be higher for untreated SCH patients compared to euthyroid controls in four primary reviews [48, 50, 53, 55], Sun et al. [43] reported a lower nonsignificant estimate (RR 0.86, 95% CI 0.56 to 1.32) [43]. A possible reason for this difference, despite very high overlap between pairs of these primary reviews, could be discrepancies in the determination of treatment status. Furthermore, Sun et al. [43] rated the quality of evidence for cardiovascular mortality in their primary review as low because of high heterogeneity.

Only one primary review considered the relationship between thyroid peroxidase antibody status and cardiovascular mortality. Collet et al. [53] found that untreated thyroid antibody-negative SCH was significantly associated with cardiovascular mortality (HR 1.34 95% CI 1.07 to 1.69), but the same did not apply for antibody-positive SCH (HR 1.28, 95% CI 0.94 to 1.72) [53], except for patients that also had TSH levels above 10mIU/L (HR 1.92, 95% CI 1.09 to 3.36) [53]. Finally, compared to euthyroid controls, death and hospitalisation due to cardiovascular causes were more likely to occur among treated SCH patients with heart failure [50].

**Table 6. Primary review findings on stroke.**

| Study | Outcome | Treatment status | Comparator | Effect estimates (95% CI) |
|---|---|---|---|---|
| Chaker et al. (2015) [52] | Stroke events | Untreated | Euthyroid | HR 0.96 (0.70–1.31) |
| | Fatal stroke | | | HR 1.27 (0.74–2.16) |
| Dhital et al. (2017) [41] | Stroke–modified Rankin scale | Untreated | Euthyroid | OR after 1 month 2.58 [1.13–5.91] |
| | | | | OR after 3 months 2.28 [1.33–3.91] |
| | Stroke–mortality after 3 months | | | OR 0.20, (0.04–1.12) |

HR–Hazard Ratio; OR–Odds Ratio.

## Stroke

There were no direct comparisons of the risk of stroke between treated and untreated patients with SCH. One primary review compared untreated individuals with SCH with euthyroid controls, and no significant difference was found in either the incidence of strokes or deaths arising from strokes [52]. Dhital et al. [41] found that functional outcomes for untreated SCH (based on the modified Rankin scale) were twice as likely to be better than those for euthyroid controls 1 and 3 months after acute ischemic stroke (Table 6).

## Fractures

No primary reviews that compared the risk or likelihood of fractures between euthyroid individuals and SCH patients that did or did not receive treatment found a significant difference (Table 7). This result was similar across various types of fractures, for example, hip fractures (HR 1.02, 95% CI 0.87 to 1.19 [45]; HR 1.10, 95% CI 0.81 to 1.50 [44]) and spine fracture (HR 1.16, 95% CI 0.66 to 2.04) [45].

## Quality of life and presence of symptoms

For the 5 studies that explicitly reported quality of life outcomes, no statistically significant differences were found between patients who did and did not receive treatment for SCH (Table 8). Similarly, thyroid-related symptoms, fatigue, mental and general well-being scores were not significantly associated with treatment status. However, Helfand [36] reported that specific subgroups–patients with TSH values greater than 10 mIU/L and those with a history of Graves' disease seemed to benefit from treatment. Graves' disease is an autoimmune thyroid disorder treated with antithyroid medication, radiotherapy or surgery [56]. Nonetheless, it is

**Table 7. Primary review findings on fractures.**

| Study | Outcome | Treatment status | Comparator | Effect estimates (95% CI) |
|---|---|---|---|---|
| Blum et al. (2015) [45] | Hip fracture | Untreated | Euthyroid | HR 1.02 (0.87–1.19) |
| | Any fracture | | | HR 1.11 (0.94–1.30) |
| | Non-spine fracture | | | HR 1.13 (0.93–1.38) |
| | Spine fracture | | | HR 1.16 (0.66–2.04) |
| Reyes Domingo et al. (2019) [38] | Fractures; adults (all >65) | Treated | Untreated | HR 1.06 (0.41–2.76) |
| Yan et al. (2016) [49] | Fractures (any) | Untreated | Euthyroid | RR 1.25 (0.85–1.84) |
| | | Treated | | RR 1.22 (0.61–2.47) |
| Wirth et al. (2014) [44] | Hip fractures | Untreated | Euthyroid | HR 1.10 (0.81–1.50) |
| | Non-spine fractures | | | HR 1.11 (0.60–2.05) |

HR–Hazard Ratio; RR–Relative Risk.

**Table 8. Primary review findings on quality of life and symptoms.**

| Study | Outcome | Treatment status | Comparator | Effect estimates (95% CI) |
|---|---|---|---|---|
| Feller et al. (2018) [17] | General QoL | Treated | Untreated | SMD -0.11 (-0.25–0.03) |
| Reyes Domingo et al. (2019) [38] | Thyroid QoL—less than 12 mo | Treated | Untreated | MD 0.0 (-2.0–2.1) |
| | Thyroid QoL—more than 12 mo | | | MD 1.0 (-1.9–3.9) |
| | | | | -0.5 (-2.2–1.3) |
| Rugge et al. (2015) [39] | Quality of life | Treated | Untreated | Multiple |
| Rugge et al. (2015) [39] | Thyroid-related symptoms | Treated | Untreated | SMD 0.01 (-0.12–0.14) |
| | Fatigue and tiredness | | | SMD -0.01 (-0.16–0.15) |
| | Depressive symptoms | | | SMD -0.10 (-0.34–0.13) |
| Helfand (2004) [36] | Symptoms | Treated | Untreated | Multiple |
| Villar et al. (2007) [40] | Symptoms, mood and quality of life | Treated | Untreated | Multiple |
| Reyes Domingo et al. (2019) [38] | Fatigue/tiredness—less than 12 mo | Treated | Untreated | MD 0.4 (-2.1–2.9) |
| | Fatigue/tiredness—more than 12 mo | | | MD -3.5 (-7.0–0.0) |
| | Mental well-being | | | Multiple |
| | Physical well-being | | | MD -0.1 (-0.3–1.0) |
| | | | | -0.1 (-0.3–1.0) |
| | General well-being | | | Multiple |

SMD–Standardised Mean difference; MD–Mean Difference.

noted that the single study that this finding was based upon was a small trial of 33 participants, all of whom had previously treated Graves' disease [36].

## Secondary outcomes

Some of the included papers (n = 4) reported on cognitive function (Table 9), which was assessed using various tools such as the Letter-Digit Coding Test and Mini-Mental State Examination. All of the primary reviews found no significant difference in cognitive function between treated and untreated groups [17, 38, 39] except Villar et al. [40]. However, this result was based on only one included study with an unclear risk of bias assessment.

## Overlap

The extent of overlap in this umbrella review is shown in Fig 2, an intersection heatmap of the calculated CCA between pairs of the 20 included primary reviews. As shown, only one of the included evidence syntheses [41] had a unique set of primary publications compared to all the other primary reviews. Overall, excluding the diagonal, the pairwise comparisons showed slight (66.7%), moderate (10%), high (2.6%) and very high (20.8%) overlap. However, it should be noted that these values were obtained with no consideration of the specific outcomes

**Table 9. Primary review findings on cognitive function.**

| Study | Outcome | Treatment status | Comparator | Effect estimates (95% CI) |
|---|---|---|---|---|
| Feller et al. (2018) [17] | Cognitive function | Treated | Untreated | Difference 1.01 (95% CI −0.56 to 2.46) |
| Reyes Domingo et al. (2019) [38] | Cognitive function | Treated | Untreated | Multiple (no difference) |
| Villar et al. (2007) [40] | Cognitive function | Treated | Untreated | MD 2.4 (0.3–4.5) |
| Rugge et al. (2015) [39] | Cognitive function | Treated | Untreated | Multiple (no difference) |

MD–Mean Difference.

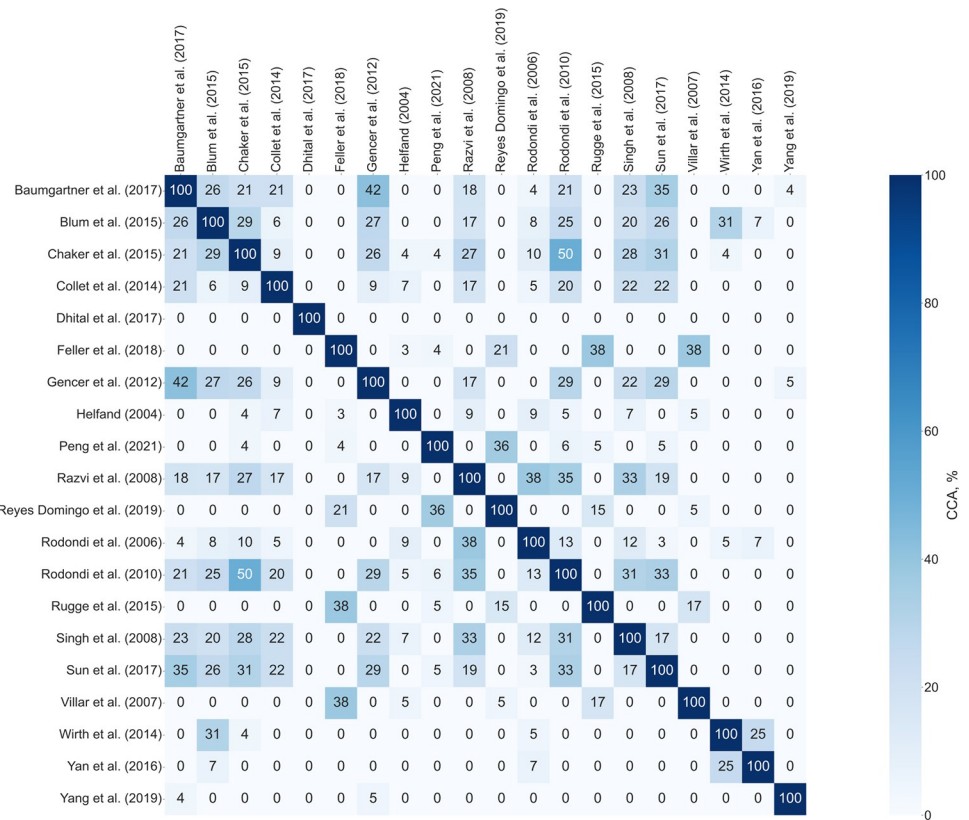

**Fig 2. Heatmap showing pairwise calculated CCA.**

reported in each of the primary reviews and therefore require cautious interpretation. That is because, although calculating CCA involved mapping all the primary studies included in each publication, not all provided findings relevant to this umbrella review. The fundamental reason for this complexity is that we were only interested in estimates reported with the participants' corresponding treatment status. On balance, CCA for the entire overview was calculated to be 5.12%, the higher limit for slight overlap [29].

## Quality appraisal

Overall confidence in review findings was found to be high for eight primary reviews [17, 38, 40, 42, 46, 47, 51, 55], moderate for seven primary reviews [39, 43–45, 52–54] and low for five primary reviews [36, 41, 48–50]. Three syntheses did not include a meta-analysis and therefore could not be assessed for questions 11, 12 and 15 [36, 38, 39]. The breakdown of checklist questions is shown in 0 10.

## Discussion

This umbrella review on the impact of the management of SCH on clinical outcomes covers evidence from 20 selected systematic reviews and meta-analyses of RCTs and observational studies. Across the outcomes of interest, the synthesised literature found can be summarised as follows. We found that the treatment of SCH may be associated with a reduced likelihood of death from all causes for patients under 70 years old. On the other hand, the relationship between SCH treatment status and the risk of death compared to the euthyroid population

Table 10. Results of the AMSTAR-2 assessments.

| Review | Question | | | | | | | | | | | | | | | | Overall confidence in results |
|---|---|---|---|---|---|---|---|---|---|---|---|---|---|---|---|---|---|
| | 1 | 2 | 3 | 4 | 5 | 6 | 7 | 8 | 9 | 10 | 11 | 12 | 13 | 14 | 15 | 16 | |
| Baumgartner et al. (2017) [51] | Y | Y | Y | Y | Y | Y | Y | Y | Y | N | Y | Y | Y | Y | Y | Y | High |
| Blum et al. (2015) [45] | Y | N | N | Y | Y | N | N | Y | Y | Y | Y | Y | Y | Y | Y | Y | Moderate |
| Chaker et al. (2015) [52] | Y | N | Y | PY | Y | Y | Y | Y | Y | N | Y | Y | N | Y | Y | Y | Moderate |
| Collet et al. (2014) [53] | Y | N | N | PY | Y | N | N | PY | N | N | Y | N | N | N | Y | Y | Moderate |
| Dhital et al. (2017) [41] | Y | N | N | PY | Y | Y | N | Y | PY | N | Y | Y | Y | N | N | Y | Low |
| Feller et al. (2018) [17] | Y | Y | Y | Y | Y | Y | Y | Y | Y | Y | Y | Y | Y | Y | N | Y | High |
| Gencer et al. (2012) [54] | Y | PY | Y | PY | Y | N | N | Y | Y | Y | Y | Y | Y | Y | Y | Y | Moderate |
| Helfand (2004) [36] | Y | PY | Y | Y | N | N | N | Y | PY | N | NMA | NMA | N | N | NMA | N | Low |
| Peng et al. (2021) [42] | Y | Y | N | PY | Y | Y | N | Y | Y | Y | Y | Y | Y | Y | Y | Y | High |
| Razvi et al. (2008) [46] | Y | N | Y | PY | Y | Y | PY | Y | Y | N | Y | Y | Y | Y | Y | Y | High |
| Reyes Domingo et al. (2019) [38] | Y | Y | Y | Y | Y | Y | Y | Y | Y | N | NMA | NMA | Y | Y | NMA | Y | High |
| Rodondi et al. (2006) [47] | Y | N | Y | N | Y | Y | Y | N | Y | N | Y | Y | Y | Y | Y | Y | High |
| Rodondi et al. (2010) [55] | Y | PY | Y | PY | Y | N | N | Y | Y | Y | Y | Y | Y | Y | Y | Y | High |
| Rugge et al. (2015) [39] | N | PY | Y | Y | Y | Y | N | N | PY | N | NMA | NMA | Y | N | NMA | Y | Moderate |
| Singh et al. (2008) [48] | Y | N | Y | PY | N | N | N | Y | N | N | Y | N | N | N | N | N | Low |
| Sun et al. (2017) [43] | Y | N | Y | PY | Y | Y | N | PY | Y | N | Y | Y | Y | Y | N | Y | Moderate |
| Villar et al. (2007) [40] | Y | Y | Y | Y | Y | Y | Y | Y | Y | Y | Y | Y | Y | Y | N | Y | High |
| Wirth et al. (2014) [44] | Y | PY | N | Y | Y | Y | Y | Y | Y | N | Y | Y | Y | Y | Y | Y | Moderate |
| Yan et al. (2016) [49] | Y | N | N | PY | Y | Y | N | Y | Y | N | Y | Y | N | Y | N | Y | Low |
| Yang et al. (2019) [50] | Y | N | N | PY | Y | Y | N | N | N | N | Y | N | N | Y | Y | Y | Low |

Y- Yes; N–No; PY–Partial Yes; NMA–No Meta-Analysis.

remains unclear. Increased risk of all-cause mortality for untreated SCH was reported in only one primary review [50], for which inclusion was restricted to patients with comorbid heart failure.

We also found that compared to euthyroidism, untreated SCH patients with very high TSH (>10mIU/L) may be at greater risk of cardiovascular events and death from cardiovascular disease [54, 55]. The same effects were observed for thyroid peroxidase antibody-negative patients [53]. Even so, there was discordance in findings between the primary reviews; whereas seven primary reviews reported a higher risk of CHD and cardiovascular mortality for untreated SCH patients than euthyroid persons [46–48, 50, 53–55], one primary review did not [43]. We rated the latter as having more than one non-critical weakness according to the AMSTAR-2 checklist; the others were either 'high' or 'moderate' in overall confidence in their results (Table 10). A high degree of overlap was calculated between the studies reporting cardiovascular outcomes, as high as 38%. Therefore, it was not easy to ascertain the precise source of the difference in results.

It was not possible to investigate the impact of treatment on the risk of stroke because the only available comparisons were of untreated SCH and euthyroidism. The finding that untreated patients had better functional outcomes one month following stroke was reported only in one low-quality study [41] and is, therefore, inconclusive. In a similar vein, there was insufficient evidence of the impact of treatment or no treatment of SCH on fracture risk. Overall confidence in the results of three out of the four primary reviews [38, 44, 45] was rated as 'high' or 'moderate'. However, none of the effect estimates was statistically significant, so it is also not possible to make conclusions on this relationship based on the quantity of evidence.

Reported findings on quality of life and the presence of symptoms between treated and untreated SCH patients were mainly of no statistical significance. As such, we cannot definitively state whether levothyroxine treatment improves or worsens these outcomes. Nonetheless, medication potentially benefits two patient groups–patients with severe SCH from the start and those that previously received treatment for autoimmune hyperthyroidism/Graves' disease [36].

The secondary outcome reported in the included primary reviews, cognitive function, was only compared between treated and untreated SCH patients. Given that the majority of findings were similar, it may be said that among patients with SCH, levothyroxine may have no significant impact on cognitive function, notwithstanding the type of assessment tool used [17, 38–40]. Crucially, however, two points must be emphasised. First, that the amount of evidence in favour of this statement is notably low, considering that few primary reviews that reported on cognitive function. Second, that only the primary outcomes were included in the literature searches, so the findings in this review cannot accurately reflect the body of evidence regarding the relationship between SCH and cognitive function.

With reference to the number of primary reviews, the volume of evidence was discernibly skewed in favour of cardiovascular outcomes (n = 13) rather than all-cause mortality (n = 7), stroke (n = 2), fractures (n = 4), quality of life (n = 5) and cognitive function (n = 4). This observation can be explained as having arisen from the umbrella review selection process, but the relatively broad inclusion criteria make it less probable. Instead, two alternatives are suggested; either that less research has been performed on the other clinical outcomes of interest or that the evidence may not have already been synthesised due to high between-study heterogeneity, for example, in outcome definitions and measurements. Additional factors, such as the comparative ease of measuring certain outcomes over others, may also influence which types of studies are performed. However, it is not possible to conclusively account for this asymmetry of evidence from this overview alone.

Generally, it cannot be ignored that most of our findings were based on empirical studies of poor quality, as reported by the authors of the primary reviews. Equally important were the critical flaws we found in the methodological quality of five of the selected primary reviews [36, 41, 48–50] consequently rated as 'low' in overall confidence in their results. Upon inspection, there was no clear boundary of review quality based on the type of empirical research that was initially selected. For example, all the syntheses that included only RCTs did not consistently get higher AMSTAR-2 ratings than those of only observational studies. As such, it can be argued that cohort studies have an essential role in filling the gap left by insufficient randomised trials on this topic.

It should be noted that there was a tendency for papers with the lowest ratings on the AMSTAR-2 checklist to have little overlap of empirical studies with other higher-rated primary reviews. This could be explained by differences in the types of outcomes reported in these syntheses; for instance, one would expect minimal overlap between fractures and cardiovascular mortality. Collectively, the reviews included in this overview had slight overlap, but as Hennesy and Johnson [31] contend, such an observation can be attributed to the breadth of the literature. This is especially true if only a small set of identical studies is shared across the included syntheses, or the overlap is highly outcome-dependent [31]. In these cases, the overall CCA would obscure the true level of overlap.

## Strengths and limitations

To the authors' knowledge, this is the first umbrella review on this topic. This overview was conducted in a systematic manner and comprehensive searches were performed to identify the

synthesised literature on the impact of the management of SCH on long-term clinical outcomes. The database searches–including grey literature, to minimise the effects of publication bias [57]–were updated in the course of the review. Screening, data extraction and quality appraisal were all done in duplicate. Furthermore, the intended aim of the umbrella review to compare the synthesised literature on this topic was achieved, even though a secondary meta-analysis was not feasible.

Nonetheless, it is crucial to consider the limitations of this review which relied exclusively on the availability, methods and quality of existing systematic reviews and meta-analyses. Of note, it was not possible to re-analyse and pool all primary review findings due to the variety of selection criteria and outcome definitions. Combining the findings of the included reviews in spite of these differences–and potential confounders–would result in biased and misleading inferences [58].

Also, an inherent limitation of the umbrella review methodology is the limited capacity to conduct detailed evaluations of empirical studies when dealing with synthesised literature. This was particularly challenging when evaluating overlap across the included reviews, as it may have been influenced by factors such as study scope and eligibility criteria. On the other hand, because this type of review was performed, it was possible to examine a wide variety of outcomes for SCH and treatment status within our specific resource constraints. The comprehensive nature of umbrella reviews has been recommended for controversial topics [19]. Furthermore, in this overview, we included IPD meta-analyses, which have been described as beneficial for analysing long-term patient outcomes [59].

Another limitation was scope mismatch between the umbrella review and the included primary reviews, for example, in cases where a selected systematic review included patients with subclinical hypo- and hyperthyroidism. This problem is commonly encountered in overviews [18], and we opted to include such papers for two key reasons. First, a preliminary literature search yielded few results with precisely the same research questions. Second, for an unresolved topic such as this, it was anticipated that the exclusion of these reviews would severely restrict this synthesis by omitting potentially relevant findings. Therefore, to limit this type of bias, inclusion in the umbrella review was based on the availability of results for which treatment status was explicitly stated.

It was also not possible to calculate overlap for the included primary reviews subdivided by their reported outcomes because most included both treated and untreated SCH patients. Consequently, assessing overlap in this way would require a detailed inspection of all their primary studies to identify the exact data sources for the respective subgroup analyses. These activities were considered to be burdensome and beyond the scope of this umbrella review, given that systematic reviews and meta-analyses were the principal units of analysis. Even so, to visualise overlap, we created a citation matrix and presented the results of the pairwise calculations, most of which were in the 'slight' band.

## Conclusion

Through this umbrella review, we systematically gathered the existing synthesised literature on the impact of the management of subclinical hypothyroidism on clinical outcomes. Our findings seem to indicate that treatment may be beneficial for SCH patients younger than 70 years due to the higher risk of all-cause mortality and cardiovascular events. In addition, untreated SCH patients with TSH levels above 10mIU/L may be at higher risk of developing cardiovascular diseases than the euthyroid population. However, more robust evidence is needed on stroke, fractures, quality of life and cognitive function in SCH. The main challenge in investigating long-term outcomes is the need for large, adequately powered and timed randomised

trials. This overview further highlights this need, given that majority of the significant findings were based on very few empirical studies often deemed to be of poor quality by the primary reviewers. Future work in observational studies may also be instrumental in strengthening the evidence base.

## Supporting information

**S1 Appendix. MEDLINE search strategy.**
(PDF)

**S2 Appendix. AMSTAR-2 checklist.**
(PDF)

**S3 Appendix. Excluded papers.**
(PDF)

**S4 Appendix. PRISMA checklist.**
(PDF)

**S5 Appendix. PRISMA abstract checklist.**
(PDF)

## Acknowledgments

We wish to thank Vicki Cormie, a senior librarian at the University of St Andrews, who assisted with the development and execution of the database searches.

## Author Contributions

**Conceptualization:** Brenda S. Bauer, Amaya Azcoaga-Lorenzo, Utkarsh Agrawal, Colin McCowan.

**Data curation:** Brenda S. Bauer.

**Formal analysis:** Brenda S. Bauer.

**Funding acquisition:** Colin McCowan.

**Investigation:** Brenda S. Bauer, Amaya Azcoaga-Lorenzo, Utkarsh Agrawal, Adeniyi Francis Fagbamigbe, Colin McCowan.

**Methodology:** Brenda S. Bauer, Amaya Azcoaga-Lorenzo.

**Project administration:** Brenda S. Bauer.

**Supervision:** Amaya Azcoaga-Lorenzo, Utkarsh Agrawal, Colin McCowan.

**Visualization:** Brenda S. Bauer.

**Writing – original draft:** Brenda S. Bauer.

**Writing – review & editing:** Brenda S. Bauer, Amaya Azcoaga-Lorenzo, Utkarsh Agrawal, Adeniyi Francis Fagbamigbe, Colin McCowan.

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
