## [Decision Letter · Decision Letter 0]

30 Mar 2022

PONE-D-22-05735The impact of the management strategies for patients with subclinical hypothyroidism on long-term clinical outcomes: An umbrella reviewPLOS ONE

Dear Dr. Bauer,

Thank you for submitting your manuscript to PLOS ONE. After careful consideration, we feel that it has merit but does not fully meet PLOS ONE’s publication criteria as it currently stands. Therefore, we invite you to submit a revised version of the manuscript that addresses the points raised during the review process.

We look forward to receiving your revised manuscript.

Kind regards,

Mahmoud Nassar, MD, PhD, MSc, MHA, MPA, CPHQ, SSBB

Academic Editor

PLOS ONE

Journal Requirements:

Reviewers' comments:

Reviewer's Responses to Questions

**Comments to the Author**

1. Is the manuscript technically sound, and do the data support the conclusions?

Reviewer #1: Yes

Reviewer #2: Yes

2. Has the statistical analysis been performed appropriately and rigorously? 

Reviewer #1: Yes

Reviewer #2: Yes

3. Have the authors made all data underlying the findings in their manuscript fully available?

Reviewer #1: Yes

Reviewer #2: Yes

4. Is the manuscript presented in an intelligible fashion and written in standard English?

Reviewer #1: Yes

Reviewer #2: Yes

5. Review Comments to the Author

Reviewer #1: Although a statistically sound and well thought out paper, the main conclusion, namely that treatment for SCH in patients less than 70 may be beneficial is based off of limited data. The paper does little to further the current knowledge base for SCH and does not change or modify the current thought process surrounding SCH. Its main conclusion was based off of one limited study and really does not warrant being made without further research into the subject. Although there is a large role for "negative studies" in the existing medical literature, a negative umbrella review does little to further critical thinking or existing treatment paradigms. It was a well thought out and reasoned paper which kept me engaged throughout, but does not warrant publication in a major journal.

Reviewer #2: Dear Author,

I appreciate your time and effort in researching this subject.

The effort and work shown here were impressive. I have some suggestions to improve it.

-Please use the PRISMA check list when preparing an abstract and manuscript (and attach them as a supplement)

Please mention the name of the databases, keywords for searching, and inclusion and exclusion criteria in the abstract.

Please specify the duration of mortality outcomes in the methods section.

-Including gray literature weakens the analysis, please clarify if any studies have been included.

Please cite the figures in your manuscript.

The included study was not statistically analyzed.

6. PLOS authors have the option to publish the peer review history of their article (what does this mean?). If published, this will include your full peer review and any attached files.

Reviewer #1: No

Reviewer #2: **Yes: **Mahmoud Nassar, MD, PhD

---

## [Author Response · Author response to Decision Letter 0]

7 Apr 2022

Comments & Responses

Reviewer #1: Although a statistically sound and well thought out paper, the main conclusion, namely that treatment for SCH in patients less than 70 may be beneficial is based off of limited data. The paper does little to further the current knowledge base for SCH and does not change or modify the current thought process surrounding SCH. Its main conclusion was based off of one limited study and really does not warrant being made without further research into the subject. Although there is a large role for "negative studies" in the existing medical literature, a negative umbrella review does little to further critical thinking or existing treatment paradigms. It was a well thought out and reasoned paper which kept me engaged throughout, but does not warrant publication in a major journal.

Response: 

We thank Reviewer #1 for this feedback. Though we appreciate their comments on the lack of novel insights generated from this tertiary review, we must emphasise that its primary aim was to compare and contrast existing systematic reviews on the impact of the management of subclinical hypothyroidism on long-term outcomes. This is unlike typical secondary reviews (i.e. systematic reviews or meta-analyses) of empirical studies, which are usually performed with the expectation of new findings arising from the synthesis of earlier studies. Evidence of this distinction for tertiary studies is provided below:

Umbrella reviews are conducted to provide an overall examination of the body of information that is available for a given topic, and to compare and contrast the results of published systematic reviews.2 The wide picture obtainable from the conduct of an umbrella review is ideal to highlight whether the evidence base around a topic is consistent or contradictory, and to explore the reasons for the findings. Furthermore, an umbrella review allows ready assessment of whether review authors addressing similar review questions independently observe similar results and arrive at generally similar conclusions. (Aromataris et al, 2015)

Source: Aromataris, E., Fernandez, R., Godfrey, C.M., Holly, C., Khalil, H. and Tungpunkom, P., 2015. Summarizing systematic reviews: methodological development, conduct and reporting of an umbrella review approach. JBI Evidence Implementation, 13(3), pp.132-140.

In light of the above explanation, we put forward that an umbrella review can neither be positive nor negative, as such a description does not align with the overall purpose of this kind of tertiary research.

Nonetheless, a statement has been added to the Strengths and Limitations section for added clarity:

“Furthermore, the intended aim of the umbrella review to compare the synthesised literature on this topic was achieved, even though a secondary meta-analysis was not feasible.” (Lines 550 - 552)

Also, the quality and availability of literature is acknowledged as a key limitation in the manuscript, including in the Conclusion:

“The main challenge in investigating long-term outcomes is the need for large, adequately powered and timed randomised trials. This overview further highlights this need, given that majority of the significant findings were based on very few empirical studies often deemed to be of poor quality by the primary reviewers. Future work in observational studies may also be instrumental in strengthening the evidence base.” (Lines 597 - 602)

Reviewer #2: Please use the PRISMA check list when preparing an abstract and manuscript (and attach them as a supplement).

Response: 

We thank Reviewer #2 for this recommendation. An additional PRISMA checklist for abstracts has been filled and is attached as Supplementary File S5 (S5 Appendix).

Reviewer #2: Please mention the name of the databases, keywords for searching, and inclusion and exclusion criteria in the abstract.

Response:

We agree with this suggestion. This section of the abstract has been revised and now reads:

“We conducted comprehensive searches on MEDLINE, EMBASE, Scopus, Web of Science, Cochrane Database of Systematic Reviews, JBI Evidence Synthesis, the PROSPERO register, Epistemonikos Database and PDQ Evidence from inception to February and July 2021 using keywords on subclinical hypothyroidism, treatment with levothyroxine, monitoring and primary outcomes (all-cause mortality, cardiovascular events, stroke, frailty fractures and quality of life). Only systematic reviews and meta-analyses on adult patient populations were considered.” (Lines 27 - 39)

Reviewer #2: Please specify the duration of mortality outcomes in the methods section.

Response:

We thank Reviewer #2 for this suggestion. The methods section has been revised to include the duration:

“…all-cause mortality, defined as the death of patients with SCH, irrespective of the cause, at least 12 months from baseline or the start of follow-up…” (Lines 177 - 178)

Reviewer #2: Including gray literature weakens the analysis, please clarify if any studies have been included.

Response:

We thank Reviewer #2 for this comment. However, umbrella review guidance recommends the inclusion of grey literature for wider coverage, thereby minimising the effects of publication bias. The main example of this is the Joanna Briggs Institute Manual for Evidence Synthesis which states:

A comprehensive search for a JBI Umbrella Review should also encompass a search for grey literature or reports that are not commercially published. As decision makers are increasingly required to base their decisions on available evidence, more and more research syntheses are being commissioned by practitioners and health care policy makers in governments globally; as a result many reports available via government or organizational websites are syntheses of research evidence and may be eligible for inclusion in a JBI Umbrella Review. A JBI Umbrella Review should include a search of at least two or three relevant sources for “grey” reports. (Aromataris et al. 2020)

Source: Aromataris E, Fernandez R, Godfrey C, Holly C, Khalil H, Tungpunkom P. Chapter 10: Umbrella Reviews. In: Aromataris E, Munn Z (Editors). JBI Manual for Evidence Synthesis. JBI, 2020. Available from https://synthesismanual.jbi.global. https://doi.org/10.46658/JBIMES-20-11

Nonetheless, the first paragraph in the Results section has been amended to show that grey literature was included:

“One item of grey literature, a systematic evidence review commissioned by a government agency for healthcare research, was included [36].” (Lines 266 - 267)

Justification for including grey literature has also been added to the Strengths and Limitations section:

“The database searches – including grey literature, to minimise the effects of publication bias [57] – were updated in the course of the review.” (Lines 547 - 549)

Reviewer #2: Please cite the figures in your manuscript.

Response:

Additional in-text citations have been included in the Results section for all the sentences where effect estimates were copied from the outcome summary tables. (Lines 315 - 403)

Reviewer #2: The included study was not statistically analyzed.

Response:

We thank Reviewer #2 for this comment. The 20 studies included, though eligible for inclusion in this umbrella review, were too varied for a meaningful and non-misleading statistical analysis. This is a particular weakness of umbrella review methodology, given the level of overlap between some of the reviews. The process of unpicking data from all the individual empirical studies was beyond the scope of this tertiary review. A statement has been added to the Strengths and Limitations to highlight this reasoning:

“Combining the findings of the included reviews in spite of these differences – and potential confounders – would result in biased and misleading inferences [57].” (Lines 557 - 558)

---

## [Decision Letter · Decision Letter 1]

22 Apr 2022

The impact of the management strategies for patients with subclinical hypothyroidism on long-term clinical outcomes: An umbrella review

PONE-D-22-05735R1

Dear Dr. Bauer,

We’re pleased to inform you that your manuscript has been judged scientifically suitable for publication and will be formally accepted for publication once it meets all outstanding technical requirements.

Kind regards,

Mahmoud Nassar, MD, PhD, MSc, MHA, MPA, CPHQ, SSBB

Academic Editor

PLOS ONE

Additional Editor Comments (optional):

Dear Author

Congratulations!

1. Please check proof of the article at least twice (best done by two different people) before finalizing.

2. Please share your study with academic colleagues, in medical conferences and in social media (LinkedIn, twitter), so that it is cited widely. If you are putting your study on twitter.

3. Please continue to contribute to the Journal.

Reviewers' comments:

Reviewer's Responses to Questions

**Comments to the Author**

1. If the authors have adequately addressed your comments raised in a previous round of review and you feel that this manuscript is now acceptable for publication, you may indicate that here to bypass the “Comments to the Author” section, enter your conflict of interest statement in the “Confidential to Editor” section, and submit your "Accept" recommendation.

Reviewer #1: All comments have been addressed

Reviewer #2: All comments have been addressed

2. Is the manuscript technically sound, and do the data support the conclusions?

Reviewer #1: Yes

Reviewer #2: Yes

3. Has the statistical analysis been performed appropriately and rigorously? 

Reviewer #1: Yes

Reviewer #2: N/A

4. Have the authors made all data underlying the findings in their manuscript fully available?

Reviewer #1: Yes

Reviewer #2: Yes

5. Is the manuscript presented in an intelligible fashion and written in standard English?

Reviewer #1: Yes

Reviewer #2: Yes

6. Review Comments to the Author

Reviewer #1: The authors have addressed all pending comments and have met criteria for submission. I recommend publication.

Reviewer #2: Dear Author

I have reviewed the article again and I'd thank you for responding to all comments.

Best regards

7. PLOS authors have the option to publish the peer review history of their article (what does this mean?). If published, this will include your full peer review and any attached files.

Reviewer #1: No

Reviewer #2: No

---

## [Editor Report · Acceptance letter]

12 May 2022

PONE-D-22-05735R1 

The impact of the management strategies for patients with subclinical hypothyroidism on long-term clinical outcomes: An umbrella review 

Dear Dr. Bauer:

I'm pleased to inform you that your manuscript has been deemed suitable for publication in PLOS ONE. Congratulations! Your manuscript is now with our production department. 

Kind regards, 

on behalf of

Dr. Mahmoud Nassar 

Academic Editor

PLOS ONE